# Integrative Analysis of the lncRNA and mRNA Transcriptome Revealed Genes and Pathways Potentially Involved in the Anther Abortion of Cotton (*Gossypium hirsutum* L.)

**DOI:** 10.3390/genes10120947

**Published:** 2019-11-20

**Authors:** Yuqing Li, Tengfei Qin, Na Dong, Chunyan Wei, Yaxin Zhang, Runrun Sun, Tao Dong, Quanjia Chen, Ruiyang Zhou, Qinglian Wang

**Affiliations:** 1Cotton Engineering Research Center of Ministry of Education, College of Agriculture, Xinjiang Agricultural University, Xinjiang Urumqi 830000, China; lyq120327@163.com (Y.L.); chqjia@126.com (Q.C.); 2Collaborative Innovation Center of Modern Biological Breeding of Henan Province, Henan Key Laboratory Molecular Ecology and Germplasm Innovation of Cotton and Wheat, School of Life Science and Technology, Henan Institute of Science and Technology, Henan Xinxiang 453003, China; qintengfeisam@163.com (T.Q.); dongna1229@163.com (N.D.); weichunyan688@163.com (C.W.); 18339511268@163.com (Y.Z.); sunrunrun123@163.com (R.S.); dongtao985@163.com (T.D.); 3Key Laboratory of Plant Genetics and Breeding, College of Agriculture, Guangxi University, Nanning 530005, China

**Keywords:** cotton, cytoplasmic male sterility, lncRNA, transcriptome, *cis*-target gene

## Abstract

Cotton plays an important role in the economy of many countries. Many studies have revealed that numerous genes and various metabolic pathways are involved in anther development. In this research, we studied the differently expressed mRNA and lncRNA during the anther development of cotton between the cytoplasmic male sterility (CMS) line, C2P5A, and the maintainer line, C2P5B, using RNA-seq analysis. We identified 17,897 known differentially expressed (DE) mRNAs, and 865 DE long noncoding RNAs (lncRNAs) that corresponded to 1172 *cis*-target genes at three stages of anther development using gene ontology (GO), Kyoto Encyclopedia of Genes and Genomes (KEGG) enrichment of DE mRNAs; and *cis*-target genes of DE lncRNAs probably involved in the degradation of tapetum cells, microspore development, pollen development, and in the differentiation, proliferation, and apoptosis of the anther cell wall in cotton. Of these DE genes, LTCONS_00105434, LTCONS_00004262, LTCONS_00126105, LTCONS_00085561, and LTCONS_00085561, correspond to *cis*-target genes Ghir_A09G011050.1, Ghir_A01G005150.1, Ghir_D05G003710.2, Ghir_A03G016640.1, and Ghir_A12G005100.1, respectively. They participate in oxidative phosphorylation, flavonoid biosynthesis, pentose and glucuronate interconversions, fatty acid biosynthesis, and MAPK signaling pathway in plants, respectively. In summary, the transcriptomic data indicated that DE lncRNAs and DE mRNAs were related to the anther development of cotton at the pollen mother cell stage, tetrad stage, and microspore stage, and abnormal expression could lead to anther abortion, resulting in male sterility of cotton.

## 1. Introduction

In recent years, with the development of sequencing technology, numerous genes and gene families have been identified. In addition, some transcriptomes, which were previously regarded as being “dark” or “junk” having either no or only a weak protein-coding ability, also have been identified [1,2,3]. However, multiple functions have recently been discovered for noncoding regions.

Long noncoding RNA (lncRNA) is considered a part of the “dark” or “junk” transcriptomes. In previous studies, lncRNA was underestimated and even considered to be transcriptional noise because of its low expression level and weak sequence conservation, when compared to mRNA [4]. Generally, lncRNAs are defined as being larger than 200 nucleotides in length, and play vital regulatory roles in biological processes in plants and animals [5,6,7], such as plant growth and development, epigenetics, and stress response [5,7]. They have important regulatory functions in organisms, and are strictly regulated at the transcriptional and post-transcriptional levels [8]. Studies have shown that lncRNAs have high stage- and tissue-specific expression patterns [9,10].

lncRNAs are categorized into three categories—long intergenic RNAs (lincRNAs), intronic ncRNAs (incRNAs), and natural antisense transcription (NATs)—based on the origin of genome [4,11] and into four categories, such as lincRNAs, incRNAs, antisense ncRNAs (ancRNAs), and sense ncRNAs (slncRNAs) based on the genomic location of lncRNAs [12].

In previous studies, lncRNAs were shown to play pivotal roles in the stress responses of plant development. In Chinese cabbage, lncRNA regulates heat stress, and 4594 putative lncRNAs were identified by strand-specific RNA-sequencing [13]. In wheat, lncRNAs were responsive to powdery mildew infection and heat stress [14]. In *ELF18-INDUCED LONG-NONCODING RNA1 (ELENA1)* mutants of a factor enhancing resistance against *Pseudomonas* in *Arabidopsis*, long noncoding RNAs were implicated in the transcriptional regulation of plant innate immunity [15]. LncRNA16397 is resistant to *Phytophthora infestans* by co-expressing glutaredoxin in tomato [16]. LncRNAs affect the root development response to cadmium stress at an early stage of rice [17]. Phosphate deficiency-induced lncRNAs (PDILs) involved in Pi-deficiency signaling and Pi transport have shown that lncRNAs regulate the responses of plants to phosphate starvation [18,19]. LncRNAs are significantly altered under nitrogen deficiency in *Populus* [20].

Some fertility-related lncRNAs have been identified in plants in recent years. In rice, we obtained 2224 reliably expressed lncRNAs, including 1624 lincRNAs and 600 lncNATs involved in rice reproduction [10]; photoperiodic-sensitive male sterility (PSMS) is regulated by lncRNA long-day-specific male-fertility-associated RNA (LDMAR) [21], as well as lncRNAs MS1T transgenic plants associated with male sterility under long-day conditions [22]. In diploid strawberry, 5884 lncRNAs were identified from the strawberry RNA-seq data during flower and fruit development [23]. In trifoliate orange, genome-wide screening showed 6584 potential lncRNAs associated with flowering development [24]. In tomato, 79,322 putative lncRNAs were identified, consisting of 70,635 lincRNAs, 8085 ancRNAs, and 602 slncRNAs related to regulation during fruit ripening [12].

The *Gossypium raimondii* (Dt), *Gossypium arboreum* (At), and *Gossypium hirsutum* (AtDt) genomes are now sequenced [25,26,27], which has promoted a huge leap forward in cotton genomics [28]. With the development of modern technology, the de novo assembly of *Gossypium hirsutum* and *Gossypium barbadense* genomes in high quality has improved contiguity and completeness in the available genome information [29,30]. In cotton, lncRNAs participate in resistance to *Verticillium dahliae* [31], drought stress [32], salt stress [33], and other abiotic stresses [34]. LncRNAs also participate in regulating fiber development [5,35] and in lint initiation [36]. However, little is known about the mechanisms of lncRNA during the cytoplasmic male sterility (CMS) in cotton; therefore, it is necessary to identify novel lncRNA and analyze the lncRNA in the development of cotton anther.

In this research, the expression of mRNAs and lncRNAs during different stages of anther development in cotton CMS line C2P5A and maintainer line C2P5B was characterized using RNA-sequencing technology. Our results demonstrate the molecular mechanisms involving lncRNAs and mRNAs during the anther development of cotton between the CMS line and the maintainer line and thus provides new perspectives for the utilization of heterosis.

## 2. Materials and Methods

### 2.1. Plant Materials

The CMS line C2P5A and maintainer line C2P5B of cotton (*Gossypium hirsutum* L.) are near-isogenic lines. Plants were cultivated in the experimental field, in Henan, China, under normal management conditions. Both lines were identified with 1.5% (w/v) acetocarmine staining and observed using scanning electron microscopy (EPSON EXPRESSION 12000XL, Nagoya, Japan).

Three different stages of anther development—the pollen mother cell stage (Pms; 3–4 mm), tetrad stage (Tds; 4.1–5.0 mm), and mononuclear stage (Ms; 5.1–6.0 mm)—and flower buds with a certain length (length from nectary to bud apex; mm) were selected for study. For pure anthers, we first peeled off the sepals and petals, discarded the pistil, and the resulting anthers were collected and placed in a centrifugal tube, which was frozen in liquid nitrogen and stored at −80 °C until use.

### 2.2. Library Construction and RNA-Sequencing (RNA-Seq)

Total RNA was extracted from the harvested anthers using an RNAprep Pure Plant Kit (Tiangen, Beijing, China) using three biological replicates per sample. The RNA concentration was detected by NanoDrop (Waltham, MA, USA), and quality was tested using an Agilent 2100 instrument (Santa Clara, CA, USA). cDNA library construction and sequencing were performed by Beijing Genomics Institute (Shenzhen, China) using the Illumina HiSeqTM 2000 system (Illumine, San Diego, CA, USA).

### 2.3. Transcript Assembly, Alignment, and Identification of Genes

In this study, raw reads were filtered to remove the adapter, low-quality reads, and contaminating sequences. Clean reads were then aligned to the reference genome (http://cotton.hzau.edu.cn/EN/download.php) by HISAT [29,37] and Bowtie2 [38], and the transcript was assembled by String Tie [39]. For mRNA identification, uniquely mapped and properly paired reads were used in the transcript construction with Cufflinks, and the constructed transcripts were compared with the cotton (*Gossypium hirsutum*) gene annotation using Cuffcompare [40].

These transcripts were compared with known mRNAs and lncRNAs in order to obtain information about their location relationships using Cufflinks, and were then merged into the transcript assembly result using Cuffmerge [40]. FPKM (Fragments PerKilobase Million) was used to calculate gene expression levels by RSEM [41].

For lncRNA identification, we assembled a single transcript meeting the following requirements: sequence length > 200 bp, FPKM > 0.5, and coverage > 1. The newly assembled transcript was compared to the cotton (*Gossypium hirsutum*) reference genome annotations so as to remove annotated transcripts (mRNA) and transcripts overlapping with other noncoding RNA species (e.g., tRNA, rRNA, snRNA, snoRNA, and miRNA) using Cuffcompare [42]. For the prediction of the transcript-coding ability, we used the Coding Potential Calculator (CPC) [43], txCdsPredict, and Coding-Non-Coding Index (CNCI) [44]—three predictive software—and a database Pfam [38]. The transcripts with a significant coding potential (CPC score > 0, CNCI score > 0, or txCdsPredict > 500) were discarded.

### 2.4. Differential mRNA and lncRNA Expression Analyses

The transcripts were aligned to the reference sequence with Bowtie2 [38], and RSEM used to calculate the expression levels of the genes and transcripts [41]. The differentially expressed genes (DEGs) were analyzed with DEGseq software [45]. Transcripts (mRNA or lncRNA) were normalized according to the |log_2_Ratio| ≥ 2, read number > 5, *p* ≤ 0.001, and FDR < 0.001. GO annotation and KEGG pathway enrichment analysis were performed for the mRNA DEGs.

### 2.5. lncRNA–mRNA Pathway Network Construction

LncRNAs can *cis*-regulate mRNAs, and it is believed that the function of lncRNA is related to its neighboring protein-coding genes [46]. We performed a location–expression analysis between mRNAs and lncRNAs in the 20 kbp of sequence representing the region closest to the identified lncRNAs. We computed Pearson’s correlation and Spearman’s correlation coefficients between each pair of lncRNA–mRNA. The lncRNA–mRNA with the most significant correlations with coefficients of >0.6 were considered potential *cis*-target genes for those lncRNAs. The potential *cis*-target genes of the lncRNAs were subjected to enrichment analysis with GO annotation and KEGG pathway. DE lncRNAs’ potential *cis*-target genes, terms, and pathways were visualized using Cytoscape v3.6 (http://www.cytoscape.org) [47].

### 2.6. Gene Expression Confirmed by Real-Time Quantification PCR (RT-qPCR)

RNA reverse transcription was conducted using TransScript^®^ One-Step gDNA Removal and cDNA Synthesis SuperMix (TransGen, Beijing, China). The random selection of several genes related to sterility by RT-qPCR was used to verify the RNA-sequencing (RNA-seq) data. Primer Premier 6.0 software was used to design specific primers (http://www.premierbiosoft.com/crm/jsp/com/pbi/crm/clientside/ProductList.jsp), which were synthesized by Sangon Biotech (Shanghai, China). The RT-qPCR reactions were performed with a qPCR SuperMix Kit (TransGen, Beijing, China). Each reaction was performed in three biological and three technical replicates on a QuantStudio 6 Flex instrument (Applied Biosystems, Foster City, CA, USA). RT-qPCR was performed according to the protocol of the TransStart^®^ Top Green qPCR SuperMix in two steps (TransGen, Beijing, China). The expression levels were quantified relative to that of the housekeeping gene *GhACT4* (GenBank accession no. AAP73451.1). The relative expression of each gene in every sample was calculated using the cycle threshold (Ct) 2^−ΔΔCt^ method [48].

## 3. Results

### 3.1. Identification and Characterization of lncRNA and mRNA

The RNA Seq data were uploaded to NCBI’s SRA database (project code: PRJNA579288). A total of 1,394,119,526 clean reads were obtained by sequencing all 18 libraries. Each library produced an average of 11.62 million data. The clean reads were then matched to the cotton reference genome by HISAT. The average genome alignment rate was 76.98% (Table 1). A total of 178,166 transcripts were detected in 18 anther tissues of cotton, and 28,047 novel lncRNAs were identified using the following four programs: CPC, txCdsPredict, CNCI, and Pfam (Figure 1A); and 46,245 novel mRNAs and 103,874 known transcripts were identified. Many lncRNAs transcripts were mainly composed of 1–7 exons, whereas mRNAs had a wide range of exons, 1–10 (Figure 1B). The transcript lengths distribution of the lncRNAs was shorter than those of the mRNAs (Figure 1C).

### 3.2. Differentially Expressed (DE) mRNAs and lncRNAs

A total of 6720, 7737, 9090 known mRNAs (Figure 2A, Appendix A) and 1689, 1657, and 2012 lncRNAs (known lncRNA 0, all novel lncRNAs; Figure 2B, Appendix A) were DEGs between the CMS line and the maintainer line in the Pms, Tds, and Ms of anther development in cotton, respectively. Also, 1082 known mRNAs of DEGs and 189 lncRNAs were shared among the three stages (Appendix A).

### 3.3. DE mRNAs Enrichment Analyses

A total of 27 GO terms and 22 KEGG enriched pathways were significantly enriched for DE mRNAs at the Pms stage between C2P5A and C2P5B (Appendix A, respectively). The top 20 terms were mostly enriched for malate synthase activity, catalytic activity, and carbohydrate metabolic process (Figure 3A and Appendix A). Moreover, the top 20 pathways were primarily enriched for several processes, such as flavonoid biosynthesis, plant hormone signal transduction, and fatty acid metabolism (Figure 3D and Appendix A).

A total of 32 GO terms and 22 KEGG-enriched pathways were significantly enriched for the DE mRNAs at the Tds stage between C2P5A and C2P5B (Appendix A, respectively). The top 20 terms of the GO were mostly enriched for oxidoreductase activity, malate synthase activity, and carbohydrate metabolic process (Figure 3B and Appendix A). Furthermore, the top 20 pathways of the KEGG were mainly enriched in starch and sucrose metabolism, plant hormone signal transduction, and flavonoid biosynthesis (Figure 3E and Appendix A).

At the Ms stage of anther development, between C2P5A and C2P5B, the DE mRNAs were significantly enriched for 66 terms (Appendix A), and the top 20 terms were mostly enriched in the carbohydrate metabolic process, oxidoreductase activity, and glucan metabolic process (Figure 3C and Appendix A). This was significantly enriched for the 20 KEGG pathway (Appendix A), and the top 20 terms mostly participated in plant hormone signal transduction, MAPK signaling pathway to the plant, flavonoid biosynthesis, and starch and sucrose metabolism (Figure 3F and Appendix A).

### 3.4. Identification and Enrichment Analyses of Cis-Target Genes of lncRNAs

To identify potential *cis*-target genes (mRNAs within a 20 kbp window upstream or downstream of the lncRNAs), Pearson’s correlation matrix was calculated, in accordance with the criteria of |Pearson’s correlation| > 0.6, 523 mRNAs, 592 mRNAs, and 690 mRNAs, corresponding to 366 lncRNAs, 397 lncRNAs, and 379 lncRNAs, were considered *cis*-target genes at the Pms stage, Tds stage, and Ms stage, respectively, and details of the chromosomal mapping of lncRNA are given in the Appendix A (Appendix A). Hierarchical clustering of DE lncRNAs and their regulated mRNA were differentially expressed at different stages between C2P5B and C2P5A (Figure 4A,B).

At the Pms stage, the GO enrichment analysis showed that 523 *cis*-target genes of lncRNAs were significantly enriched for 10 processes, such as NADH dehydrogenase (ubiquinone) activity, oxidoreductase activity, and cytoplasm (Figure 5A and Appendix A). The *cis*-target genes of these lncRNA were significantly enriched for the 12 KEGG pathways, and the pathways mainly included fatty acid biosynthesis, flavonoid biosynthesis, glutathione metabolism, MAPK signaling pathway to the plant, oxidative phosphorylation, and ubiquinone and other terpenoid quinone biosynthesis (Figure 6A and Appendix A).

At the Tds stage, 592 *cis*-target genes of lncRNAs were significantly enriched for NADH dehydrogenase (ubiquinone) activity, cytoplasm, and pectinesterase activity (Figure 5B and Appendix A). Of these *cis*-target genes, they were significantly enriched for 11 KEGG pathways, which mainly consisted of glutathione metabolism, MAPK signaling pathway to the plant, oxidative phosphorylation, pentose and glucuronate interconversions, and plant hormone signal transduction (Figure 6B and Appendix A).

At the Ms stage, 690 *cis*-target genes of lncRNAs were significantly enriched for 12 GO terms and 10 KEGG pathways, and the GO terms mainly included the cytoplasm and GDP-fucose transmembrane transporter activity (Figure 5C and Appendix A). The significant KEGG pathways mainly consisted of fatty acid biosynthesis, MAPK signaling pathway to the plant, oxidative phosphorylation, pentose and glucuronate interconversions, and phenylalanine metabolism (Figure 6C and Appendix A).

In this paper, the GO annotation and KEGG pathway enrichment were performed for DE mRNAs and DE lncRNAs–*cis*-target genes. The results show that these terms and pathways play a crucial role in response to the cotton anther abortion of the CMS line C2P5A.

### 3.5. Validation of RNA-Sequencing (RNA-Seq) by Real-Time Quantitative PCR (RT-qPCR)

RT-qPCR verified the expression level of the selected DE lncRNAs and the *cis*-target of the lncRNAs. In this paper, we randomly selected three, five, and four lncRNAs, corresponding to three, five, and five *cis*-target genes of lncRNAs at the Pms, Tds, and Ms stage of the anther development, respectively, between the CMS line C2P5A and the maintainer line C2P5B (Figure 7 and Appendix A). The specific primers were designed by Primer Premier 6.0 software (Appendix A). Pearson’s coefficient was used to analyze the correlation between the qPCR and RNA-seq data. Overall, the expression level of 12 lncRNA DEGs and 13 *cis*-target genes of lncRNAs RT-qPCR was calculated using the 2^−ΔΔCt^ method, which is basically consistent with the RNA-seq data (lncRNA, Appendix A, correlation coefficient = 0.77; *cis*-target genes of lncRNAs, Appendix A, correlation coefficient = 0.58). The lncRNAs at the Pms, Tds, and Ms stage of the anther development for the RT-qPCR and RNA-seq correlation coefficient were 0.78, 0.72, and 0.92, respectively (Appendix A). The *cis*-target gene of lncRNA, at the Pms, Tds, and Ms stage of the anther development RT-qPCR and RNA-seq correlation coefficient were 0.70, 0.91, and 0.63, respectively (Appendix A). The results showed that our RNA-seq data were reliable and conducive for screening DEGs during anther development.

## 4. Discussion

### 4.1. Transcriptome mRNA in the Anther Development of Cotton

Anther development is a complex process, with numerous genes and various metabolic pathways being involved. In model plants like *Arabidopsis* and rice, many genes that regulate the fate of somatic germ cells and the differentiation of the anther wall, as well as control the degradation of tapetum cells and microspore development during anther development, have been identified [49,50,51]. In cotton, the *GhACS1* gene encodes an acyl-CoA synthetase, which is essential for normal microspore development, and highly expressed in sporogenous cells, pollen mother cells, microspores, and tapetum cells [52]. The actin-depolymerizing factor (*GhADF7*) gene may play an important role in pollen development and germination, and its transcript expression reaches a peak at flowering [53]. *GhMYB24* encodes the MYB-like transcription factor that regulates the development of the tapetum [54].

Carbohydrates provide energy and nutrients for anther development; the sink strength of anthers is the highest in the early stages of anther development, which is intensively energy-demanding [55] and, thus, abnormal carbohydrate metabolism can significantly damage pollen development and cause male sterility [56,57]. Sugar is converted to starch so as to ensure energy preservation for pollen maturation and bud germination [57]. Zhang and his team studied the sterility of the male-sterile line 1355A of cotton, and found soluble sugar and fatty acid metabolism to play a central role in anther development. In the male sterile line 1355A, soluble sugars are decreased, and fatty acid synthesis is key for regulating normal pollen hydration and the primary component of sporopollenin, which can protect pollen from various stresses and is crucial for pollen grain development and male sterility [58,59]. High rates of glucose metabolism may promote fatty acid synthesis in order to promote the anther development of cotton [59]. Researchers studying cotton (*Gossypium hirsutum*) anthers under a high temperature (HT) and normal temperature (NT) indicated that HT disturbs sugar and ROS metabolism by disrupting DNA methylation, leading to microspore sterility [60]. Flavonoids play an important role in the formation of pollen exine formation, pollen germination, and the fertility of several plants that are key branch-point genes during tetrad and uninucleate microspore periods. Some researchers who studied genic male sterility (GMS) mutant anther development indicated that flavonoid metabolism is initially activated at the tetrad stage, then suppressed at the uninucleate microspore stage, leading to male sterility and the absence of flavonoids in mature stamens [61]. By using digital gene expression (DGE), some researchers identified many of the key genes that are required for cotton anther development and those that are mainly associated with sucrose and starch metabolism, the pentose phosphate pathway, glycolysis, and flavonoid metabolism [61,62].

Hormone signal transduction plays an important role in anther development, such as that involving ethylene, gibberellic acid (GA), and abscisic acid (ABA); higher amounts of ethylene may directly lead to the premature degeneration of the tapetal layer in GMS mutant anthers [59]. GA can accelerate flowering and promote the development of female flowers; conversely, GA deficiency can cause the abnormal development of anthers [63,64]. In addition, high levels of ABA delayed endosperm differentiation and the lack of endosperm leads to difficulties in germination [65]. Some researchers have shown that the MAPK signaling pathway plays a key role in the differentiation, proliferation, and apoptosis of cells [66].

In this research, we identified 17,897 known DE mRNAs at three stages of anther development, between the CMS line C2P5A and the maintainer line C2P5B (Appendix A). These DE mRNAs underwent GO enrichment and KEGG enriched pathway analyses, which showed that the significant terms were mostly enriched in malate synthase activity, catalytic activity, carbohydrate metabolic process, oxidoreductase activity, oxygen carrier activity, and glucan metabolic process (Figure 3A–C and Appendix A), and the significantly enriched pathways were flavonoid biosynthesis, plant hormone signal transduction, fatty acid metabolism, starch and sucrose metabolism, and MAPK signaling pathways in plants (Figure 3D–F and Appendix A). Compared to the same anther developmental stage of the maintainer line C2P5B, there were many key genes with an abnormal expression pattern in the CMS line C2P5A, which indicated that the abortion of cotton C2P5A was related to the abnormal metabolic pathway of the abnormally expressed gene regulating anther development.

### 4.2. lncRNA in the Anther Development of Cotton and Predicted Functions

The genomes of the eukaryotes are universally transcribed. Some RNAs are encoded into proteins, and other thousands of lncRNAs regulate key molecular and biological processes [46]. Until now, some of the best-studied mammalian lncRNAs have associated the dysregulation of lncRNAs with reproduction, including germ cell specification, early embryo implantation and development, and reproductive hormone regulation [67,68], but the involvement of plant lncRNAs in reproduction is still poorly understood. In this study, we identified that the expression profiles of mRNA and lncRNA were related to cotton anther male sterility and were differentially expressed at different anther development periods. When further analyzed, the interaction network between the lncRNAs and mRNAs based on expression profiles shows that these transcriptomes may play crucial roles in the anther development of cotton.

In a previous study, we used a chromatin state map approach and RNA-seq to identify what is typically co-expressed with lncRNAs and involved in regulating their neighboring mRNA [69,70]. In animals, many studies have shown that lncRNAs directly regulate their neighboring genes in a *cis*-acting manner, such as *lncRNA–TCONS_00175604 cis*-action in dairy goat [71], lncRNA-NEF *cis*-regulating neighbor gene *FOXA2* in mice [72], lncRNA-Six1 *cis*-regulating neighbor gene *Six1* in chicken [73], lncRNA-Malat1 in mouse [74], lncRNA–TBILA *cis*-regulating *HGAL* in humans [75], and lncRNA–Jpx via both *trans*- and *cis*-Xist expression in mice [76].

In plants, lncRNAs regulate many molecular functions and biological processes in various ways. LncRNAs can pair with *cis*- or *trans*-transcripts, translation inhibition, and gene silencing [46]. The *Arabidopsis* noncoding RNA HID1 promotes photomorphogenesis in continuous red light (CR) and acts through PIF3 [77]. During vernalization in *Arabidopsis*, antisense lncRNA *COOLAIR* is associated with the *FLOWERING LOCUS C* (*FLC*) locus and switches of chromatin states during epigenetic regulation; *COOLAIR* participates in the autonomous pathway and controls the flowering time [8,78,79]; and intronic sense lncRNA *COLDAIR* acts as a scaffold RNA to recruit the PRC2 complex and establish *FLC* epigenetic silencing, and mediates FRIGIDA (FRI) degradation [8,80,81].

Natural antisense lncRNAs *TWISTED LEAF*(*TL*) play a *cis*-regulatory role in *OsMYB60* expression and for maintaining leaf blade flattening [82]. Overexpressing lncRNA *LAIR* upregulates the expression of the neighbor gene LRK (leucine-rich repeat receptor kinase) cluster, which increases rice grain yield [83]. LncRNA1459 was knocked out by CRISPR/Cas9 altered tomato fruit in ripening; in these mutants, the ethylene production and lycopene accumulation were largely repressed [84].

In cotton, previous studies have shown that lncRNAs XLOC_063105-CotAD_37096 and XLOC_115463-CotAD_12502 probably function in *cis*-regulating responses to drought stress [32]. One lncRNA, TCONS_00061835-Gh_D06G1439 (GhMYB-like), regulates cotton fiber development [36] by complementary base pairing with the protein-coding gene, lncRNA, and the circRNA complex network, demonstrating that some lncRNAs are involved in biotic and abiotic stresses [31,33,34].

In this study, we sought the *cis*-target genes within a radius of 20 kbp for the lncRNAs, and identified 865 lncRNAs that might exert their functions through the predicted *cis*-target genes’ 1172 mRNAs (Appendix A). The predicted *cis*-target genes underwent GO enrichment and KEGG pathway analyses, and the results provided ideas for future research. The significant GO terms were enriched in NADH dehydrogenase (ubiquinone) activity, oxidoreductase activity, photosynthetic electron transport, cytoplasm, and pectinesterase activity (Figure 5A–C). NADH dehydrogenase (ubiquinone), oxidoreductase, and photosynthetic electron transport play a crucial role in energy production and conversion, in which the proteins are components in the mitochondrial respiratory chain. This showed that mitochondrial respiratory-related enzymes play an important role in the regulation of the CMS line C2P5A, and this result is consistent with a previous study in CMS plants [85,86,87]. Pectinesterase is a wall-degrading enzyme, and abnormal expression may lead to the disruption of the cell wall structure in the CMS line C2P5A; a previous study demonstrated that pectinesterase plays a major role in the plant [88]. In our research, the GO of the cytoplasm was significant, showing that its abnormality may be one of the key factors leading to abortion in the CMS line C2P5A compared to the maintainer line C2P5B.

Significant KEGG pathways were primarily enriched for several processes, such as fatty acid biosynthesis, flavonoid biosynthesis, glutathione metabolism, MAPK signaling pathway to the plant, oxidative phosphorylation, ubiquinone and other terpenoid quinone biosynthesis, pentose and glucuronate interconversions, and plant hormone signal transduction (Figure 6A–C). Previous studies show that these metabolism pathways play a vital role during anther development in flowering plants, in which a disturbed metabolism pathway seriously leads to the impairment of anther development, and causes male sterility [55,57,61,62,85,89]. This study showed that some lncRNAs and mRNAs might play important roles in anther development, such as *cis*-target gene Ghir_A09G011050.1 of LTCONS_00105434 through GO:0008137 (NADH dehydrogenase (ubiquinone) activity) and ko00190 (oxidative phosphorylation); *cis*-target gene Ghir_A01G005150.1 of LTCONS_00004262 through GO:0016491 (oxidoreductase activity) and ko00941 (flavonoid biosynthesis); *cis*-target gene Ghir_D05G003710.2 of LTCONS_00126105 through GO:0005975 (carbohydrate metabolic process) and ko00040 (pentose and glucuronate interconversions); *cis*-target gene Ghir_A03G016640.1 of LTCONS_00085561 through GO:0016790 (catalytic activity) and ko00061 (fatty acid biosynthesis); and *cis*-target gene Ghir_A12G005100.1 of LTCONS_00085561 through GO:0005524 (ATP binding) and ko04016 (MAPK signaling pathway to the plant).

## 5. Conclusions

In this study, we carried out transcriptome studies to identify DE mRNA and lncRNA and performed GO annotation and pathway enrichment analysis on the potential *cis*-target genes of these DE lncRNAs, showing their important roles in the regulation of anther development in cotton. Of these DE genes, LTCONS_00105434, LTCONS_00004262, LTCONS_00126105, LTCONS_00085561, and LTCONS_00085561 correspond to the *cis*-target genes Ghir_A09G011050.1, Ghir_A01G005150.1, Ghir_D05G003710.2, Ghir_A03G016640.1, and Ghir_A12G005100.1, respectively, which participate in anther development. This research provides us with a better perspective of the molecular regulation of the anther development of CMS line C2P5A in cotton.

## Figures and Tables

**Figure 1 genes-10-00947-f001:**
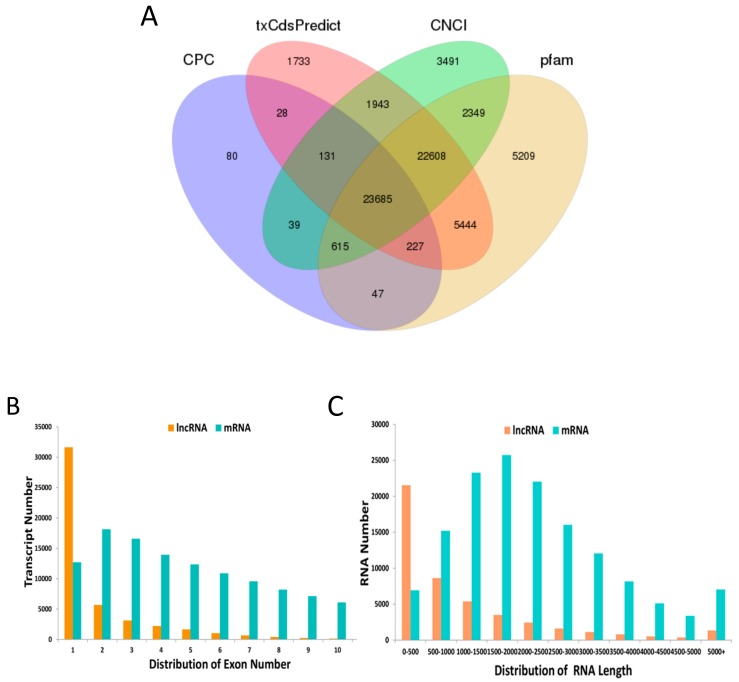
Long noncoding RNA (lncRNA) and mRNA characterization. (**A**) Venn diagram for screening results of the lncRNA by four software (Coding Potential Calculator (CPC), txCdsPredict, Coding-Non-Coding Index (CNCI), and Pfam). (**B**) The distribution of the lncRNA and mRNA exon number. (**C**) The distribution of lncRNA and mRNA length.

**Figure 2 genes-10-00947-f002:**
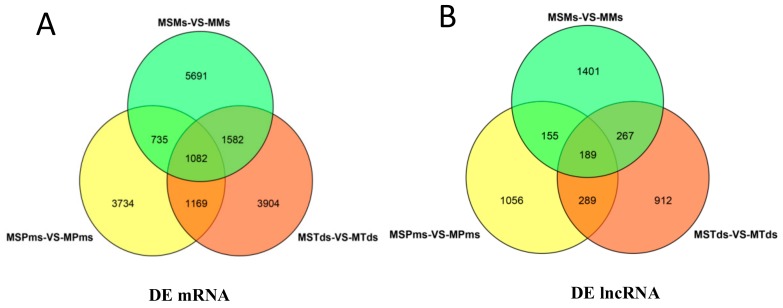
Venn diagram showing known mRNAs corresponding to differentially expressed genes (DEGs), and novel lncRNAs differentially expressed at three stages of anther development, between the cytoplasmic male sterility line C2P5A and the maintainer line C2P5B. (**A**) Venn diagram of known mRNAs of DEGs. (**B**) Venn diagram of novel lncRNA DEGs. MS—Male sterility line; M—Maintainer line; Pms—pollen mother cell stage; Tds—tetrad stage; Ms—mononuclear stage.

**Figure 3 genes-10-00947-f003:**
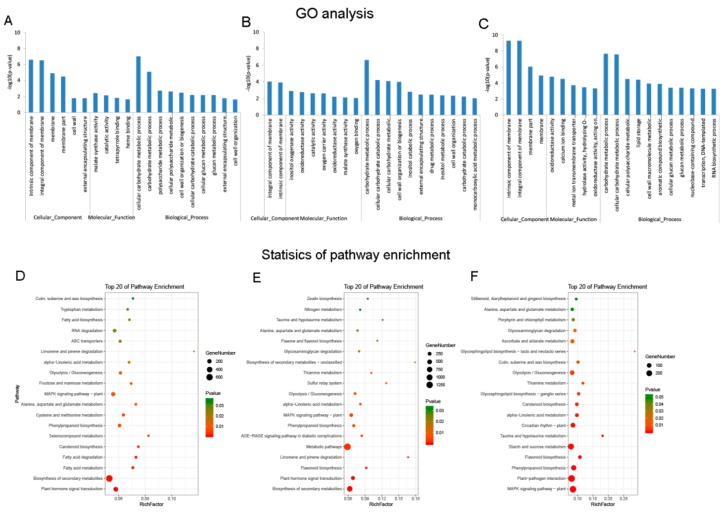
Differential expression of mRNAs at three stages of anther development, between the cytoplasmic male sterility line C2P5A and the maintainer line C2P5B. Gene ontology (GO) analysis of DEGs in (**A**) MSPms-VS-MPms, (**B**) MSTds-VS-MTds, and (**C**) MSMs-VS-MMs. Statistical KEGG enrichment of DEGs in (**D**) MSPms-VS-MPms, (**E**) MSTds-VS-MTds, and (**F**) MSMs-VS-MMs. The dot size indicates the number of enriched differentially expressed genes in each pathway, and the different color of the dot represents the *p*-value of each pathway. The top 20 enriched GO terms and KEGG enrichments ranked by *p*-values are shown. MS—Male sterile line; M—Maintainer line; Pms—pollen mother cell stage; Tds—tetrad stage; Ms—mononuclear stage.

**Figure 4 genes-10-00947-f004:**
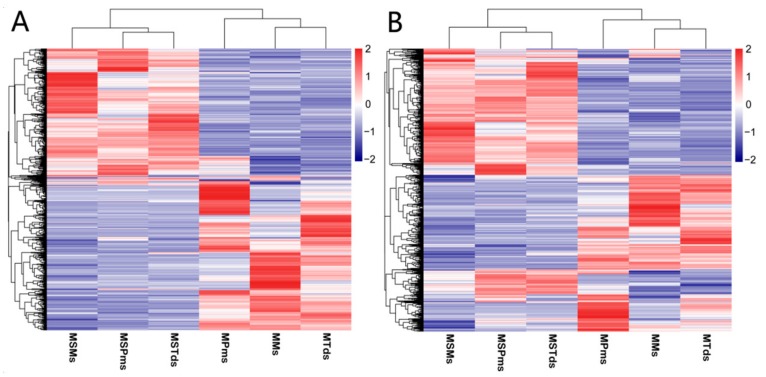
(**A**) Hierarchical clustering of lncRNAs differentially expressed at different stages in the development of cotton. (**B**) Hierarchical clustering of *cis*- target genes of lncRNAs differentially expressed at different stages in cotton development. Data are expressed as FPKM. Red: relatively high expression; Blue: relatively low expression. The bar code on the right represents the color scale of the log_2_ values. MS—Male sterile line; M—Maintainer line; Pms—pollen mother cell stage; Tds—tetrad stage; Ms—mononuclear stage.

**Figure 5 genes-10-00947-f005:**
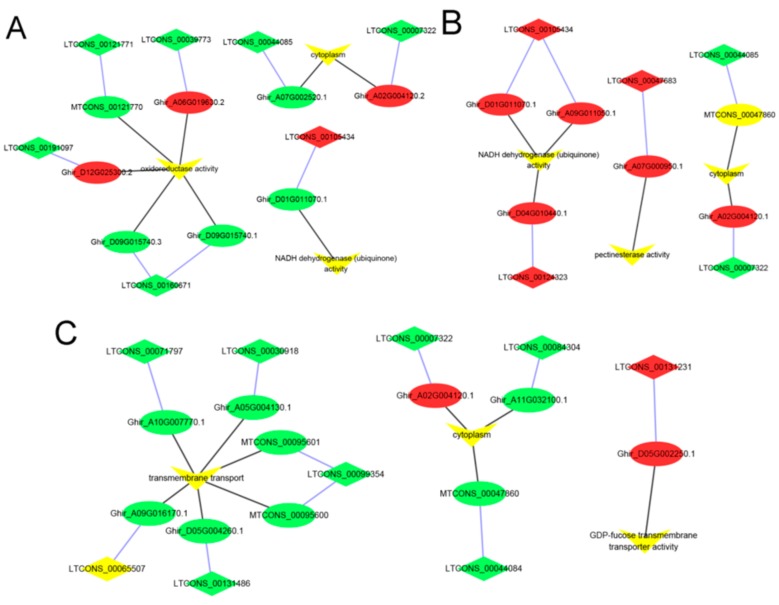
GO analysis enriched for lncRNA–potential *cis*-target genes of anther development between the cytoplasmic male sterility line C2P5A, and the maintainer line C2P5B. (**A**) GO terms enriched for the Pms stage lncRNAs–mRNAs. (**B**) GO terms enriched for the Tds stage lncRNAs–mRNAs. (**C**) GO terms enriched for the Ms stage lncRNAs–mRNAs. The ellipse, diamond, and inverted triangle nodes represent the mRNAs (*cis*-target genes), lncRNAs, and GO terms, respectively. Red represents upregulated, green represents downregulated, and yellow represents the enrichment of the GO terms. The blue edges and black edges show regulatory interactions among *cis*-target genes and lncRNAs, genes (*cis*-target genes and lncRNAs) and GO terms, respectively.

**Figure 6 genes-10-00947-f006:**
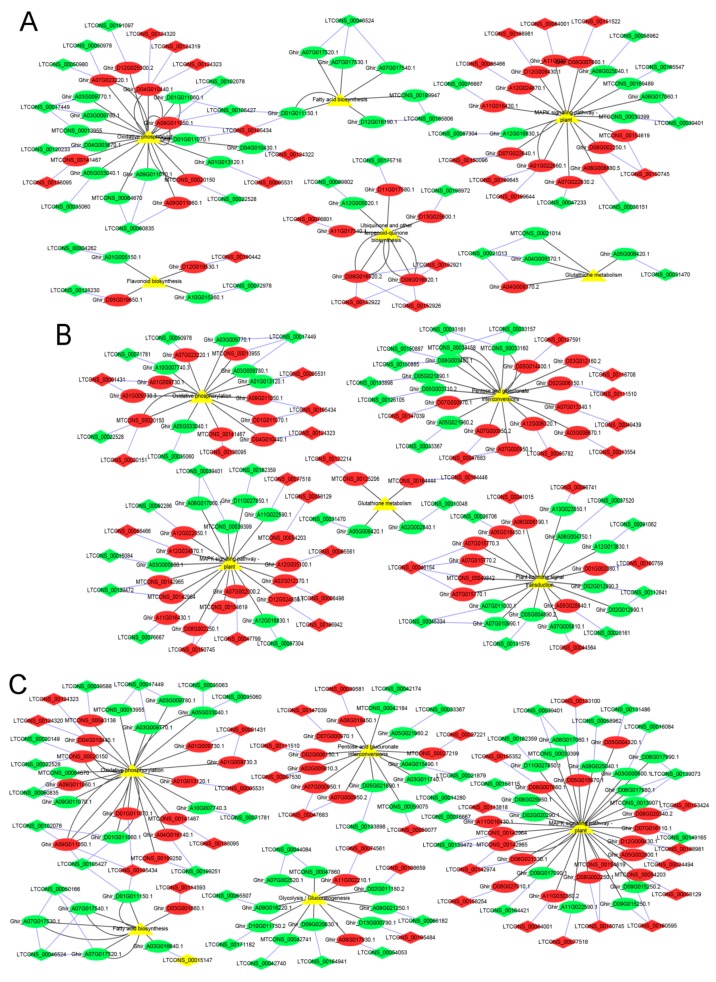
Kyoto Encyclopedia of Genes and Genomes pathways (KEGG) enriched for lncRNA–potential *cis*-target genes of anther development, between the cytoplasmic male sterility line C2P5A, and the maintainer line C2P5B. (**A**) KEGG pathways enriched for the Pms stage lncRNAs–mRNAs. (**B**) KEGG pathways enriched for the Tds stage lncRNAs–mRNAs. (**C**) KEGG pathways enriched for the Ms stage lncRNAs–mRNAs. The ellipse, diamond, and triangle nodes represent the mRNAs (*cis*-target genes), lncRNAs, and pathway, respectively. Red represents upregulated, green represents downregulated, and yellow represents the enrichment of pathways. The blue edges showed interaction between the mRNA and lncRNA. The black edges showed regulatory interaction transcriptomes and pathways.

**Figure 7 genes-10-00947-f007:**
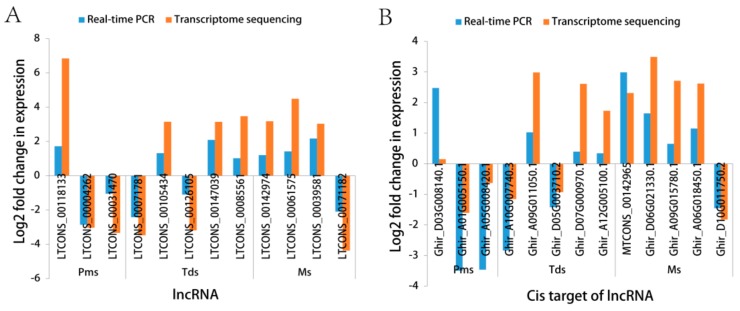
Quantitative PCR (RT-qPCR) confirmation of the RNA-sequencing (RNA-Seq) expression profiles at different stages of anther development of cotton, between the cytoplasmic male sterility line C2P5A and the maintainer line C2P5B. (**A**) The expression levels of 12 differentially expressed long noncoding RNAs. (**B**) The expression levels of 13 *cis*-target genes of lncRNAs corresponding to (**A**) lncRNAs, respectively. Of these lncRNAs, *LTCONS_00142974* regulated two *cis*-target genes, *MTCONS_00142965, Ghir_D06G021330.1*. *GhACT4*-actin was used as a reference gene for normalization in our experiments. The fold change from the RT-qPCR was calculated using the 2^−ΔΔCt^ method.

**Table 1 genes-10-00947-t001:** Summary of tag numbers.

Sample	Raw Data	Clean Data	GC (%)	Q30 (%)	Clean Reads Rate (%)	novel_lncRNA.isoforms	novel_mRNA.isoforms	known_mRNA.isoforms
MsPms1	96339898	84112026	44.33	97.65	87.308	26669	22642	77144
MsPms2	96339060	84728466	44.43	97.84	87.948	26079	22080	74534
MsPms3	96340692	84822750	44.52	97.81	88.045	26723	22570	76486
MsTds1	66418638	60016542	44.22	98.33	90.361	25569	22130	74937
MsTds2	96341884	84562392	43.57	97.68	87.773	27125	23083	78797
MsTds3	96340878	84830410	44	97.77	88.052	27515	22581	76691
MsMs1	70226770	62112146	45.21	98.35	88.445	24684	21119	72145
MsMs2	96340984	84269862	44.5	97.81	87.47	27175	22017	75181
MsMs3	75363530	68729552	44.43	98.45	91.197	26502	22011	75483
MPms1	78357630	71140510	43.98	98.42	90.79	26798	22389	74464
MPms2	93592588	83300884	43.81	97.97	89.004	27279	22445	74560
MPms3	94122576	81326894	44.24	97.83	86.405	27314	22397	74869
MTds1	93071774	84922682	43.79	98.38	91.244	27246	22535	74405
MTds2	79502370	72818484	43.85	98.23	91.593	26317	22163	73329
MTds3	71210292	64700084	43.83	98.33	90.858	26046	21985	72553
MMs1	96340778	84992800	44.36	98.03	88.221	26255	22543	75378
MMs2	96340324	84734142	43.65	97.83	87.953	26319	22745	75915
MMs3	74569750	67998900	43.6	98.23	91.188	25285	22181	73375

MS, Male sterile line C2P5A; M, Maintainer line C2P5B; 1, 2, 3 represent the biological duplications; Pms, pollen mother cell stage; Tds, tetrad stage; Ms, mononuclear stage.

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
