# Peer review of "Integrative Analysis of the lncRNA and mRNA Transcriptome Revealed Genes and Pathways Potentially Involved in the Anther Abortion of Cotton (Gossypium hirsutum L.)"

_genes, 2019, doi:10.3390/genes10120947_

Round 1

Reviewer 1 Report

The manuscript “ Integrative analysis of lncRNA and mRNA transcriptome revealed genes and pathways potentially involved in anther abortion of cotton (Gossypium hirsutum L.)” had identified the differently expressed mRNA and lncRNA during anther development of cotton between the cytoplasmic male sterility (CMS) line C2P5A and the maintainer line C2P5B using RNA seq analysis.

I appreciate the authors effort and time in performing the analysis. However, I strongly recommend rewriting and English editing for the manuscript. Most of places the authors are unclear in their explanation which makes difficulty in understanding. Introduction needs to be rewritten. The authors claim previous studies have shown that lncRNAs play major roles in stress and plant development. However they have failed to discuss about the reports on cotton. 

Line no 15: Cotton plays an important role of economics in many countries.

Please rephrase the sentence to Cotton plays an important role in economics for many countries.

Line no: 19: We identified 17,897 known differentially expression (DE) mRNAs and DE 865 lncRNAs….

Please rephrase this sentence to We identified 17,897 known differentially expression (DE)

 mRNAs and 865 DE lncRNAs.

Lines 35-39: The authors need to rephrase or remove the sentences which makes it difficult to understand the exact meaning.

Line 48-52: According to the origin of genome, lncRNAs are mainly three categories.

This paragraph needs a clear rewriting. LncRNAs are categorized into three categories as long intergenic RNAs (lincRNAs), intronic ncRNAs (incRNAs) and natural antisense transcription (NATs) based on the origin of genome and into four categories such as lincRNAs, incRNAs, antisense ncRNAs (ancRNAs) and sense ncRNAs (slncRNAs) based on genomic location of lncRNA.

Lines 81-88:  These lines needed to be rewritten clearly as

In this research, the expression characters of mRNAs, lncRNAs during different stages of the anther development in cotton CMS line C2P5A and maintainer line C2P5Bwas performed using RNA-Sequencing technology.Our results demonstrate the molecular mechanism of lncRNAs and mRNAs during the anther development of cotton between the CMS line and the maintainer line. Thus, providing us a new perspective for the utilization of heterosis.

Lines 96-100: Please rephrase:

Three different stages of the anther development, pollen mother cell stage (Pms,  3-4mm), tetrad stage (Tds, 4.1-5.0mm), mononuclear stage (Ms, 5.1-6.0mm), flower buds with length (length from nectary to bud apex, mm) were selected for the studies.  The pure anthers from peel off sepals, petals and discard pistil were collected and conserved in free centrifugal tube which was frozen in liquid nitrogen and stored at −80 °C until use.

Data availability: The authors state they have carried out RNA Seq analysis in the manuscript. However, the data availability of the RNASeq data of the CMS line and other line has not been shared or cited which neede to be added. Also, the authors claim they have identified all novel lncRNA. This needs to be compared with the previously published lncRNAs.

Materials and Methods: Needs to be re written. The authors have used broken sentences in many places. Lines 180 the authors claim the parameters used for identifying the lncRNAs. This needs to justified with previous records.

The authors could have carried out chromosomal mapping of the lncRNAs.

Also, in line 283 the authors talk about the experimental validation of the lncRNA- Overall, the expression level of 12 lncRNA DEGs and 13 cis-target of lncRNAs which needed to be justified for choosing.

Author Response

Dear Reviewer:

Thank you for your comments concerning our manuscript entitled “Integrative analysis of lncRNA and mRNA transcriptome revealed genes and pathways potentially involved in anther abortion of cotton (Gossypium hirsutum L.)” (genes-616677). Those comments are all valuable and very helpful for revising and improving our paper, as well as the important guiding significance to our researches. We have studied comments carefully and have made correction which we hope meet with approval. All problems you raised we have revised in the manuscript.Please see the attachment.

Reviewer 2 Report

Yuqing Li et al., are sequenced and did comparative analysis of potential long non-coding RNAs and mRNA in Cytoplasmic male sterility line (C2P5A) and the control line (C2P5B). Based on differential expression of LncRNA and mRNAs in three different stages, authors discussed about enriched pathways and selective genes related to anther development.

 I highly recommend authors for editing English grammar throughout the manuscript including abstract. Correct the in complete sentence in line 17-19. Line 24: is it “anther cell wall” or “anther wall cell” ? It’s very difficult to understand the method of sample collection and library construction (eg:- Replicates). Authors must use “,”, “and” at respective places for better understanding. In addition, are samples sequenced with Single- or Paired-end?. Author must give the evidence of sequencing by submitting all RNA-seq libraries to NCBI SRA or other standard public database. Line 109: Change “matched” to “mapped/aligned”. Why did you choose FPKM instead widely used TPM normalization? In Table 1: How does the mapping rate is calculated?. How did you achieve 0.7811 mapping rate for sample MsMs1, since only 50.02% of reads are mapped? Authors must give more detail of transcript assembly and mapping (against transcripts) statistics as main table since DE analysis of LncRNAs and mRNAs are based on the transcripts instead reference genome. In addition, author should give details of comprehensive comparative results of assembled transcripts and reference genome in manuscript to known and novel mRNA. Its very confusing, author said that they have mapped the assembled transcripts to reference genome and then genes was used for DEG analysis in 2.4 section (Line:130-132). Further they have used transcripts (mRNAs and LncRNAs) for finding significant DEs.  Here, what’s the importance of mapping transcripts to reference genome? I recommend to avoid the first paragraph (line: 303-310) in Discussion section or move it to introduction section if it appropriate. Line 281 : Author must write clearly about what they want to say. Author should give evidence in heatmap by showing how differentially expressed LncRNAs and their regulated mRNA between control and male sterility lines. ? How do authors claim 28,047 novel lncRNAs without comparing to other know cotton lncRNAs in Line 164.  Author should provide the details of novel lncRNAs by comparing other known lncRNAs in cotton. 

Author Response

Dear Reviewer:

Thank you for your comments concerning our manuscript entitled “Integrative analysis of lncRNA and mRNA transcriptome revealed genes and pathways potentially involved in anther abortion of cotton (Gossypium hirsutum L.)” (genes-616677). Those comments are all valuable and very helpful for revising and improving our paper, as well as the important guiding significance to our researches. We have studied comments carefully and have made correction which we hope meet with approval. All problems you raised we have revised in the manuscript. Please see the attachment.

Reviewer 3 Report

In this manuscript by Li et al., used RNAseq analysis to compare an isogenic CMS and maintainer lines of Cotton at three different stages of anther development. Differentially expressed long non-coding (Lnc) RNAs and mRNAs were analysed to Using gene ontology (GO) enirchment analysis and KEGG pathway mapping. Authors suggest these differentially expressed lncRNAs and mRNAs are responsible for anther abortion in the male sterile lines. The manuscript provides valuable information in understanding the molecular basis for male sterility. The work in this manuscript is done methodically. However, the draft needs significant improvements in the way the results are presented. Some of the figures provided in the draft are not legible. Number of grammatical errors are observed in the draft. The grammatical errors and language used in presenting the results needs significant improvements. Despite these lackings I recommed the manuscript for publication after some these revisions.

L15: "important role of economics"?

L35: "numerous of the genome is transcribed"?

L303: "important role of economics in many nationals"

L440: "we known that multiple genes"?

...are some of the first sentences used in the abstract, intro, discussion and conclusion. number of these errors needs to be corrected to present the results of the study in a better way

L103: "The RNA concentration detection by Nanodrop."...... Incomplete sentences

L277: rephrase

Fig. 6: Y axis "Fode change"...fold change

Author Response

(The authors gave the same response as above.)

Round 2

Reviewer 1 Report

Dear Authors,

The manuscript entitled “Integrative analysis of lncRNA and mRNA transcriptome revealed genes and pathways potentially involved in anther abortion of cotton (Gossypium hirsutum L.)” have discussed the role of DE mRNA and LncRNA in anther developmental process.

The authors have well addressed the concerns raised earlier. However I still recommend the following changes to be amended.

Line no 21: Gene ontology (GO) and the Kyoto Encyclopedia of Genes and Genomes (KEGG) enrichment of DE mRNAs, and cis-target genes of DE lncRNAs confirmed their crucial roles, such as in the degradation of tapetum cells;, microspore development;, pollen development;, and the differentiation, proliferation, and apoptosis of the anther cell wall in cotton.

Please rephrase to: Gene ontology (GO), Kyoto Encyclopedia of Genes and Genomes (KEGG) enrichment of DE mRNAs, and cis-target genes of DE lncRNAs confirmed their crucial roles in the degradation of tapetum cells;, microspore development;, pollen development;, in the differentiation, proliferation, and apoptosis of the anther cell wall in cotton.

Line 25: Of these DE genes, LTCONS_00105434, LTCONS_00004262, LTCONS_00126105, LTCONS_00085561, and LTCONS_00085561, corresponding to cis-target genes Ghir_A09G011050.1, Ghir_A01G005150.1, Ghir_D05G003710.2, Ghir_A03G016640.1, and Ghir_A12G005100.1, respectively, participated in oxidative phosphorylation, flavonoid biosynthesis, pentose and glucuronate interconversions, fatty acid biosynthesis , and MAPK signaling pathway in –plants, respectively. In summary, the transcriptomic data elucidated that DE lncRNAs and DE mRNAs regulate the anther development of cotton at the pollen mother cell stage, tetrad  stage, and microspore stage, and thate abnormal expression led to anthers abortion, resulting in the male sterility of cotton.

Please rephrase to: Of these DE genes, LTCONS_00105434, LTCONS_00004262, LTCONS_00126105, LTCONS_00085561, and LTCONS_00085561, correspond to cis-target genes Ghir_A09G011050.1, Ghir_A01G005150.1, Ghir_D05G003710.2, Ghir_A03G016640.1, and Ghir_A12G005100.1, respectively. They participated in oxidative phosphorylation, flavonoid biosynthesis, pentose and glucuronate interconversions, fatty acid biosynthesis , and MAPK signaling pathway in –plants, respectively. In summary, the transcriptomic data elucidated that DE lncRNAs and DE mRNAs regulate the anther development of cotton at the pollen mother cell stage, tetrad  stage, microspore stage, and the abnormal expression led to anthers abortion, resulting in the male sterility of cotton.

Introduction:

Line 38: In recent years, with the development of sequencing technology numerous genes have been found to be transcribed; thus, some transcriptomes which were previously regarded as being “dark” or “junk” have either no or only a weak protein-coding ability, [1-3].

To be rephrased as “In recent years, with the development of sequencing technology numerous genes and gene families have been identified; also some transcriptomes which were previously regarded as being “dark” or “junk” having either no or only a weak protein-coding ability also have been identified [1-3].

Materials and Methods:

Line 114: In this study, the raw reads were filtered to remove the adapter, low-quality reads, and contaminating sequences. The clean reads were aligned to the reference genome (http://cotton.hzau.edu.cn/EN/download.php) by HISAT [29, 37] and Bowtie2 [38], and the transcript was assembled by String Tie [39]. For the mRNA identification, uniquely mapped and properly paired reads were used in the transcript construction with Cufflinks, and the constructed transcripts were compared with the cotton (Gossypium hirsutum) gene annotation using Cuffcompare [40].

Please rephrase to” In this study, raw reads were filtered to remove adapter, low-quality reads, and contaminating sequences. Clean reads were then aligned to the reference genome (http://cotton.hzau.edu.cn/EN/download.php) by HISAT [29, 37] and Bowtie2 [38], and the transcript was assembled by String Tie [39]. For mRNA identification, uniquely mapped and properly paired reads were used in the transcript construction with Cufflinks, and the constructed transcripts were compared with the cotton (Gossypium hirsutum) gene annotation using Cuffcompare [40].”

To avoid the extensive usage of “the” from the sentence.

Result and Discussion

Line 189:  The differentially expressed genes (DEGs) were analyzed with software DEGseq [43], in accordance with the threshold |log2Ratio| ≥ 2, read number > 5, p ≤ 0.001, and the criteria of FDR < 0.001;, a total of 6720, 7737, 9090 known mRNAs (Figure 2A, Supplementary Table S1a, b, c) and 1689, 1657, and 2012 lncRNAs (known lncRNA 0, all novel lncRNAs; ) (Figure2B, Supplementary Table S2a, b, c.) were DEGs between the CMS line and the maintainer line in the Pms, Tds, and Ms of anther development in cotton, respectively. Also, 1082 known mRNAs of DEGs and 189 lncRNAs were shared among the three stages (Supplementary Table S1d,S2d).

Please remove the repetitive materials and methods information: to “A total of 6720, 7737, 9090 known mRNAs (Figure 2A, Supplementary Table S1a, b, c) and 1689, 1657, and 2012 lncRNAs (known lncRNA 0, all novel lncRNAs; ) (Figure2B, Supplementary Table S2a, b, c.) were DEGs between the CMS line and the maintainer line in the Pms, Tds, and Ms of anther development in cotton, respectively. Also, 1082 known mRNAs of DEGs and 189 lncRNAs were shared among the three stages (Supplementary Table S1d,S2d).

”Conclusion:” Line 479 :In this study, based on molecular biology analyses, we found DE mRNAs and performed  GO annotation and pathway enrichment analysis on the potential cis-target genes of these DE  lncRNAs, which confirmed their important roles in the regulation of anther development in  cotton. Of these DE genes, LTCONS_00105434, LTCONS_00004262, LTCONS_00126105, LTCONS_00085561, and LTCONS_00085561 correspond to the cis-target genes  Ghir_A09G011050.1, Ghir_A01G005150.1, Ghir_D05G003710.2, Ghir_A03G016640.1, and  Ghir_A12G005100.1, respectively, which participate in anther development. This research provides us with a better perspective of the molecular regulation of the anther development  of CMS line C2P5A in cotton”.

Please include –“In this study we carried our RNAseq analysis/transcriptome studies to identify the DE mRNA and lncRNA and performed performed  GO annotation and pathway enrichment analysis on the potential cis-target genes of these DE  lncRNAs, which…”

Thank you

Author Response

Dear Reviewer:

Thank you for your comments concerning our manuscript entitled “Integrative analysis of lncRNA and mRNA transcriptome revealed genes and pathways potentially involved in anther abortion of cotton (Gossypium hirsutum L.)” (genes-616677). Those comments are all valuable and very helpful for revising and improving our manuscript, as well as the important guiding significance to our researches. We have studied comments carefully and have made correction according to your comments. All problems you raised we have revised in the manuscript.The main corrections in the paper and the responds to your comments see the attachment.

We deeply appreciate you for your good comments and warm work earnestly, and hope that the correction will meet with approval.

Once again, thank you very much for your comments and suggestions.

Reviewer 3 Report

In the earlier review of the manuscript, I recommended the manuscript for publication provided improvements in the language used in the manuscript. Authors corrected and improved the language used in the manuscript and addressed relevant queries. I recommend the draft for publication. 

Few minor corrections found in the manuscript are enlisted below:

L 16: "another"...to "anther" development

L109-112: Fix errors in the the sentence, probably split the sentence into two.

Figure 4: Increase the font size of the sample labels provided below the hierarchial clustering

L499: "appreciated"...to "appreciate"

Author Response

Dear Reviewer:

Thank you for your comments concerning our manuscript entitled “Integrative analysis of lncRNA and mRNA transcriptome revealed genes and pathways potentially involved in anther abortion of cotton (Gossypium hirsutum L.)” (genes-616677). Those comments are all valuable and very helpful for revising and improving our manuscript, as well as the important guiding significance to our researches. We have studied comments carefully and have made correction according to your comments. All problems you raised we have revised in the manuscript.The main corrections in the paper and the responds to your comments see the attachment.

We deeply appreciate you for your good comments and warm work earnestly, and hope that the correction will meet with approval.

Thank you and best regards.

This manuscript is a resubmission of an earlier submission. The following is a list of the peer review reports and author responses from that submission.

Round 1

Reviewer 1 Report

Please provide more information on the CMS line and near isogenic lines?

Is there any linkage map available for the CMS charecters to know the responsible genes? If so identify genes in that locus from the available genome sequence and look for differential expression of those genes.

All the numbers are wrong more than 1 billion reads are obtained from 18 replicates and average number of reads cont be 11 millions.

what is the reference genome is it hirsutum? how many total annotated genes available?

How many genes are identified in cotton from the reference genome? How can you have 178,166 transcripts? this applies for all other numbers, Please give how many genes are detected in total. 

table1: mapping rate to percentage conversion required, give expressed genes number for all the reps.

Figure 4 and 5 should not be in manuscript, can be moved to supplementary info. what is the value for this analysis? Are there any important conclusions made out of this

overall English improvement is needed, it is hard to understand conclusions.